# The Role of rs713041 Glutathione Peroxidase 4 (*GPX4*) Single Nucleotide Polymorphism on Disease Susceptibility in Humans: A Systematic Review and Meta-Analysis

**DOI:** 10.3390/ijms232415762

**Published:** 2022-12-12

**Authors:** Priscila Barbosa, Nada F. Abo El-Magd, John Hesketh, Giovanna Bermano

**Affiliations:** 1Centre for Obesity Research and Education (CORE), School of Pharmacy and Life Sciences, Robert Gordon University, Sir Ian Wood Building, Garthdee Road, Aberdeen AB10 7GJ, UK; 2Biochemistry Department, Faculty of Pharmacy, Mansoura University, Mansoura 35516, Egypt

**Keywords:** genetic polymorphism, glutathione peroxidase 4, human disease, meta-analysis

## Abstract

Aim: The single-nucleotide polymorphism (SNP) rs713041, located in the regulatory region, is required to incorporate selenium into the selenoprotein glutathione peroxidase 4 (GPX4) and has been found to have functional consequences. This systematic review aimed to conduct a meta-analysis to determine whether there is an association between *GPX4* (rs713041) SNP and the risk of diseases in humans and its correlation with selenium status. Material and methods: A systematic search for English-language manuscripts published between January 1990 and November 2022 was carried out using six databases: CINAHL, Cochrane, Medline, PubMed, Scopus and Web of Science. Odds ratios (ORs) and 95% confidence intervals (CIs) were applied to assess a relationship between *GPX4* (rs713041) SNP and the risk of different diseases based on three genetic models. Review Manager 5.4 and Comprehensive Meta-Analysis 4 software were used to perform the meta-analysis and carry out Egger’s test for publication bias. Results: Data from 21 articles were included in the systematic review. Diseases were clustered according to the physiological system affected to understand better the role of *GPX4* (rs713041) SNP in developing different diseases. Carriers of the *GPX4* (rs173041) T allele were associated with an increased risk of developing colorectal cancer in additive and dominant models (*p* = 0.02 and *p* = 0.004, respectively). In addition, carriers of the T allele were associated with an increased risk of developing stroke and hypertension in the additive, dominant and recessive models (*p* = 0.002, *p* = 0.004 and *p* = 0.01, respectively). On the other hand, the *GPX4* (rs713041) T allele was associated with a decreased risk of developing pre-eclampsia in the additive, dominant and recessive models (*p* < 0.0001, *p* = 0.002 and *p* = 0.0005, respectively). Moreover, selenium levels presented lower mean values in cancer patients relative to control groups (SMD = −0.39 µg/L; 95% CI: −0.64, −0.14; *p* = 0.002, I^2^ = 85%). Conclusion: *GPX4* (rs713041) T allele may influence colorectal cancer risk, stroke, hypertension and pre-eclampsia. In addition, low selenium levels may play a role in the increased risk of cancer.

## 1. Introduction

The dietary intake of selenium is essential for health. Severe deficiency, in combination with virus infection, was found to cause myocarditis in China, but more recently, various studies have indicated that a sub-optimal intake can increase the risk of diseases such as cancer [1,2]. Glutathione peroxidase 4 or phospholipid hydroperoxide glutathione peroxidase (GPX4) was one of the first selenium-containing proteins, selenoproteins, discovered over 40 years ago [3]. Unlike other glutathione peroxidase family members, GPX4 has the unique ability to reduce hydroperoxides in complex lipids such as phospholipid, cholesterol and cholesterolester hydroperoxides, even when they are inserted into biomembranes or lipoproteins [4] and, therefore, to prevent iron (Fe2+)-dependent formation of toxic lipid reactive oxygen species (ROS) [5].

GPX4 is vital for normal physiology, and its absence is incompatible with life due to its role in preserving mitochondrial function, inflammation, differentiation, immunity and cell death [3]. Inhibition of GPX4 function leads to lipid peroxidation and can induce ferroptosis, an iron-dependent, non-apoptotic form of cell death [6]. Key to the function and activity of GPX4 is the trace element selenium and its incorporation into its catalytic center as selenocysteine (Sec) residue [7]. Sec is a selenium-containing amino acid encoded by the UGA codon, usually known as a stop codon. The discovery of how the in-frame UGA codon is differentiated from a UGA stop codon signaling has led to a detailed understanding of how selenium, as Sec, is incorporated into selenoproteins and its regulation. A special recognition sequence in the 3′-untranslated region (UTR) called selenocysteine insertion sequence (SECIS) allows the Sec incorporation to the GPX4 through a specific tRNA [8]. The SECIS element is an RNA that adopts a stem-loop structural motif, which directs the cells to translate UGA codons as Sec [8,9].

Dietary selenium influences the level of selenoproteins to different extents depending on the tissue and the specific selenoproteins. For example, low dietary selenium affects glutathione peroxidase 1 and selenoprotein W more than GPX4 [10], and it has been proposed that this leads to a hierarchy in selenoprotein synthesis where the various selenoproteins are differentially affected by dietary selenium supply [11]. It has been proposed that the SECIS-based incorporation mechanism and 3′UTR sequences are the basis for this hierarchy [11]. Given the importance of 3′UTR sequences in selenoprotein synthesis, it has been proposed that genetic variation in the 3′UTR of selenoprotein genes could affect the expression of such proteins.

A single-nucleotide polymorphism (SNP), rs713041 or GPX4c718t, has been found at position 718 of the *GPX4* mRNA in the 3′UTR, the regulatory region required for incorporation of selenium into selenoproteins [12]. Rs713041 was shown to be functional in regulating GPX4 synthesis, modulating the synthesis of GPX4 by altering the affinity of the Sec insertion machinery for its SECIS element and protein associated with the 3′UTR [12]. Additional work has shown that over-expression of transcripts containing the *GPX4* 3′UTR with either the T or C allele altered selenoprotein expression and hierarchy in selenium-depleted transfected cells, suggesting the importance of selenium supply in modulating the impact of *GPX4* (rs713041) SNP on redox and antioxidant cell functions [13].

According to the SNP database of the National Library of Medicine, the C allele of *GPX4* (rs713041) SNP is the frequent allele reported by various studies and populations [14]. Substitution of the major C allele into the minor T allele has been reported to be linked to different types of cancer, such as breast cancer [15,16,17], colorectal cancer [18], lung and laryngeal cancer [19] and prostate cancer [20]. Moreover, other diseases have been reported to be linked with *GPX4* (rs713041) SNP, such as obesity [21], endometriosis [22], autoimmune thyroid diseases [23], Alzheimer’s disease [24], depression [25] and multiple sclerosis [26]. The C allele of *GPX4* (rs713041) SNP was shown to have a protective role against oxidative DNA damage when selenium levels are adequate in in-vitro studies [13]; in addition, to maintain GPX4 concentrations in lymphocytes from individuals with CC genotype better than in individual with TT genotype when selenium intake falls [12].

Due to the vital role of GPX4 in biological functions, the regulation of its expression closely linked to selenium status, and the potential relationship between selenium status and disease risk, it is crucial to investigate the importance of the *GPX4* (rs713041) SNP in modulating the susceptibility to various human diseases. To date, the various studies investigating the influence of *GPX4* (rs713041) on disease risk have been small and, therefore, to further evaluate this relationship, a systematic review with meta-analysis was carried out to assess the association between *GPX4* (rs713041) SNP and risk of diseases in three genetic models and its correlation with selenium status.

## 2. Materials and Methods

### 2.1. Study Design

The systematic review protocol used follows the recommendations of the Preferred Reporting Items for Systematic Review and Meta-analysis Protocol (PRISMA-P) [27], and to ensure the quality of the protocol, the PRISMA-P checklist was completed (Appendix A). The acronym PICOS, which corresponds to P = patient or population, I = intervention or indicator, C = comparison or control, O = outcome, and S = Study, was used to define the guiding question for this study [28]. In this systematic review and meta-analysis, the acronym PICOS corresponds to P = patients with various diseases; I = *GPX4* (rs713041) SNP genotype; C = people without disease; O = susceptibility to disease; and S = case-control studies.

### 2.2. Search Strategy

A systematic search of English language manuscripts published between January 1990 and November 2022 was made using six databases: PubMed, Medline, Web of Science, Cochrane, Scopus and CINAHL. The following keywords were used in each database: *GPX4,* Glutathione peroxidase 4, Phospholipid glutathione peroxidase, *PH-GSH*, *GPX-4,* Polymorphism*, rs713041, GPx4 T/C 718, SNP, genetic, variant, SNP, single nucleotide polymorphism, mutation. The exact strings used for each database are reported in Appendix A.

### 2.3. Inclusion and Exclusion Criteria

This review considered reports that examined the association between *GPX4* (rs713041) SNP and susceptibility to human diseases. As a prerequisite for inclusion, only case-control studies in humans were considered. To be included, papers needed to be published in a peer-reviewed journal and to report *GPX4* (rs713041) SNP genotype in samples obtained from patients confirmed by the diagnostic criteria for each disease type as part of a case-control study. Articles were also required to present original data, supply sufficient information on genotype frequencies of the *GPX4* (rs713041) SNP, provide allelic and genotypic frequencies in case and control groups, and report distributions of genotype frequencies in the control group that were within the Hardy-Weinberg equilibrium (HWE) range. If studies provided overlapping data, the papers with the largest number of participants were chosen.

Reports were excluded if they did not satisfy the specified inclusion criteria or if the results were published in specific publication types, such as letters, abstracts, reviews, meta-analyses and proceedings, and unpublished sources of data and studies. Papers without extractable numerical data, with non-reliable data or with data reported more than once in the literature were excluded.

### 2.4. Data Extraction

To ensure consistency in reviewing and reporting results, two independent authors (P.B. and N.F.A.) carried out the data extraction. Disagreement between both researchers was resolved by consensus. The following information was extracted for each paper: first author’s name, year of publication, study country, ethnicity and gender of patients, disease type, sample size (cases/controls), genotyping methods, number of each genotype and HWE.

### 2.5. Quality Assessment

Two investigators (N.F.A. and G.B.) independently performed a quality assessment of the included study. Disagreement was resolved by discussion. Quality assessment was accomplished by evaluating bias using the Newcastle-Ottawa quality assessment scale for case-control studies [29]. Quality was assessed in three categories: selection, comparability and exposure. In the selection category, a maximum of four stars could be allotted in the function of the definition of cases, representativeness of the cases, selection of controls and definition of controls. In the comparability category, a maximum of two stars could be assigned based on the comparability of cases and controls on the basis of the design or analysis. In the exposure category, a maximum of three stars could be allocated based on the ascertainment of exposure, the same method for cases and controls and the non-response rate. Studies that scored >8 were classified as high quality, 4–7 as moderate quality and <3 of poor quality [29].

### 2.6. Data Analysis

The association between the risk of diseases and *GPX4* (rs713041) SNP was estimated using odds ratio (OR) and 95% of confidence intervals (CIs). ORs and 95% CIs of TT versus CC (additive model), CT+TT versus CC (dominant model) and TT versus CT+CC (recessive model) were calculated. The data were analyzed using fixed effects (Mantel-Haenszel) and random effects models (DerSimonian and Laird). Random effects were more appropriate when heterogeneity between studies was present. The I^2^ test was used to evaluate the heterogeneity of the studies. If I^2^ > 50%, the results were defined as heterogeneous. Chi-squared tests were used to assess the variation across the studies, and *p* < 0.05 was considered statistically significant. The fixed-effects model was used to calculate ORs when the studies were homogeneous (I^2^ < 50%). To better understand the influence of *GPX4* (rs713041) SNP on the development of different diseases, diseases were clustered according to body systems affected to perform the meta-analysis. Sensitivity analysis was also performed to assess the influence of individual studies on OR and 95% CI by excluding each study in turn. Sub-group analysis was also conducted to assess the association of *GPX4* (rs713041) SNP with a specific type of cancer or a subset of diseases.

The potential publication bias was examined visually in a funnel plot of log[OR] against its standard error (SE), and the degree of asymmetry was tested by Egger’s test (*p* < 0.05 was considered a significant publication bias [30]). When publication bias was present, the Duval and Tweedie’s trim and fill method was used to determine where missing studies are likely to fall and then recalculate the combined effect [31].

The risk of disease as a function of selenium levels and GPX3 activity (a commonly used marker of selenium status) in plasma/serum was also evaluated by continuous analysis, using mean and standard deviation for both selenium levels and GPX3 activity, reported in some of the studies. The standardized mean difference was used for this meta-analysis as the studies assessed selenium levels and measured GPX3 activity using different methods, allowing expression of the effect size between cases and controls relative to the variability observed in each study [32].

Given the diversity of diseases considered in the selected studies, it was not possible to carry out the meta-analysis for all diseases. For meta-analysis purposes, according to Cochrane guidelines, at least two reports are required to perform statistical analysis [33]. The software Review Manager, Version 5.4, was used to perform the meta-analysis, whereas the software Comprehensive Meta-Analysis, Version 4, was used to carry out Egger’s test for publication bias and the trim and fill effect.

## 3. Results and Discussion

### 3.1. Description of Included Studies

The database searches identified 889 potentially eligible reports. Zotero software was used [34] to find duplicate publications, and 446 duplicated reports were excluded; therefore, 443 reports were further screened. Titles and abstracts of all the papers were screened for relevance, and 387 reports were excluded as irrelevant to the aim of this study. The full text was retrieved for all reports apart from one. The remaining 55 reports were systematically reviewed for further details. After full-text reading, 19 reports were excluded for not being case-control studies, 6 for not being related to *GPX4* (rs713041) SNP and 9 for other reasons (Figure 1). Finally, 21 reports were included in the analysis for this systematic review.

The selected studies reported data related to *GPX4 *(rs713041) SNP linked to nineteen diseases; these included breast cancer [16,17], colorectal cancer [35,36], prostate cancer [20,37], laryngeal cancer [19], lung cancer [19], pre-eclampsia [38,39], hypertension [40], ischemic stroke [41], endometriosis [22], recurrent miscarriage [42], pregnancy loss [43], Graves’ disease [23], Hashimoto disease [23], acute pancreatitis [44], Alzheimer’s disease [24], depression [25], multiple sclerosis [26], type 2 diabetes mellitus [45] and Kashin-Beck disease [46]. Regarding demographics, from the twenty-one reports, one study was performed on Arabs [45], two reports on Russians [42,43], one report on Caucasian Americans [36], one report on South Americans [24], five reports on Asians, Han Chinese population [22,23,38,39,46], and the other eleven reports on Europeans [16,17,19,20,25,26,35,37,40,41,44]. Genotyping was carried out by different methods, two reports using real-time PCR [19], one KASPar [35], two reports MassArray [23,37], four reports PCR-RFPL [42,43,44,46] and the remaining twelve reports using TaqMan assay [16,17,20,22,24,25,26,36,38,39,40,41,45]. The characteristics, genotyping methods, sample size and genotype frequency of the twenty one reports are presented in Table 1.

Evaluation using the Newcastle-Ottawa quality assessment scale for case-control studies revealed that ten selected studies were of high quality (three studies scored 9, seven studies scored 8), whereas twelve were of moderate quality (score = 7), as indicated in Table 1. More detailed scoring for each study can be found in Appendix A.

To better assess the influence of *GPX4* (rs713041) SNP on the development of different diseases, it was necessary to cluster diseases in groups according to body systems affected or disease characteristics. Three distinct groups with fourteen reports were created to evaluate the influence of *GPX4* (rs713041) SNP in susceptibility to cancer, hypertension-related diseases, and reproduction.

Selected studies reported data related to selenium levels in plasma or serum for only four types of cancer: breast cancer [17], prostate cancer [20,37], laryngeal cancer [19] and lung cancer [19] and all studies were carried out in Europeans. Graphite furnace atomic absorption spectrometry (GFAAS) was used to determine selenium levels in three studies [17,19], whereas dynamic reaction cell-inductively coupled plasma field mass spectrometry [37] and a fluorometric method [20] were used for the two remaining studies. Two studies [17,37] reported levels of GPX3 activity measured by using methods based on Paglia and Valentine [47]. The levels of selenium and GPX3 activity and sample size of the five reports are presented in Table 2.

### 3.2. Meta-Analysis Cancer

Seven reports regarding cancer were included: two for breast cancer [16,17], one that evaluated laryngeal and lung cancer [19], two for prostate cancer [20,37] and two for colorectal cancer [35,36]. Genotypes were available for 3260 cases, and 3883 controls, and the results were incorporated into the meta-analysis. Only the study by Peters et al. [36] was performed on Caucasian Americans; the other six studies were performed on a European population [16,17,19,20,35,37]. Regarding genotyping, four studies used TaqMan technology to analyze SNP [16,17,20,36], one used real-time PCR [19], one used KASPar [35], and one used MassArray [37]. The meta-analysis results for all types of cancer are reported in Figure 2.

As shown in Figure 2, no between-study heterogeneity was found in overall comparisons in the three genetic models (I^2^ < 50%). Using a fixed-effects model and all studies pooled into the meta-analysis, *GPX4* (rs713041) SNP was not statistically related to cancer (Figure 2: A—additive model: OR, 1.09; 95% CI, 0.96–1.25; *p* = 0.20; B—dominant model: OR, 1.09; 95% CI, 0.99–1.21; *p* = 0.08; C—recessive model: OR, 1.04; 95% CI, 0.92–1.17; *p* = 0.54). Sensitivity analysis was performed to determine whether any study had a greater degree of influence between the association of *GPX4* (rs713041) SNP and cancer risk. No single study had a larger influence over the other studies when assessing the association for the additive or recessive model. However, in the dominant model, a positive association was found between the rs713041 T allele and cancer risk when data for laryngeal cancer from Jaworska et al. study [19] were removed (OR, 1.11; 95% CI, 1.00–1.23; *p* = 0.05), or when data for lung cancer from the same study were removed (OR, 1.11; 95% CI, 1.01–1.23; *p* = 0.04). No asymmetry was noted in the resultant funnel plots for the additive and recessive model (Appendix A) and supported by Egger’s test (additive model *p* = 0.17, recessive model *p* = 0.45), suggesting the lack of publication bias; whereas publication bias was present in the dominant model (*p* = 0.04, Appendix A).

To assess the influence of *GPX4* (rs713041) SNP on the susceptibility of breast, colorectal and prostate cancer, separately, sub-group analysis was carried out, following Cochrane guidelines that define a minimum of two reports for meta-analysis [33]. Including only papers related to one type of cancer, no statistically significant association was observed between *GPX4* (rs713041) SNP and breast or prostate cancer using a fixed-effects model and for each genetic model (Table 3). However, when studies related to colorectal cancer were analyzed separately, the meta-analysis showed that carriers of the *GPX4* (rs713041) T allele were associated with an increased risk of developing colorectal cancer in comparison to those with the C allele in the additive model (OR, 1.28; 95% CI, 1.04–1.58; *p* = 0.02) and the dominant model (OR, 1.25; 95% CI, 1.07–1.46; *p* = 0.004) (Table 3).

### 3.3. Meta-Analysis Hypertension-Related Diseases

The association of *GPX4* (rs713041) SNP with hypertension-related diseases was also tested. Four reports were included in this group: two evaluated pre-eclampsia [38,39], one evaluated hypertension [40] and one evaluated ischemic stroke [41]. A total of 2149 cases and 2250 controls were incorporated into the meta-analysis for hypertension-related diseases. The two studies evaluating pre-eclampsia were performed on the Chinese Han population [38,39], whereas the other two were on a European population [40,41]. All the reports used TaqMan technology for genotyping the SNP [38,39,40,41].

The results are presented in Figure 3. All three genetic models had an I^2^ higher than 50% (I^2^ = 90%, 80% and 89%, respectively), indicating the presence of heterogeneity between the studies. For this reason, a random-effects model was used to investigate the association of rs713041 with hypertension-related diseases. Results show that *GPX4* (rs713041) SNP did not increase the risk for hypertension-related diseases (Figure 3: A—additive model: OR, 1.60; 95% CI, 0.75–3.39; *p* = 0.22; B—dominant model: OR, 1.09; 95% CI, 0.75–1.59; *p* = 0.64; C—recessive model: OR, 1.49; 95% CI, 0.79–2.83; *p* = 0.22). Sensitivity analysis did not show any study having a greater degree of influence between the association of *GPX4* (rs713041) SNP and risk of hypertension-related diseases. Publication bias was evident in the visual analysis of funnel plots (Appendix A) and supported by Egger’s test (additive model *p* = 0.004; dominant model *p* = 0.035; recessive model *p* = 0.0005). The trim and fill analysis identified one possible missing study. When the funnel distribution was rebalanced by including this putative additional study (Appendix A), the adjustment for publication bias produced a negligible effect on the pooled estimates (additive model: OR, 1.10; 95% CI, 0.51–2.37; dominant model: OR, 0.93; 95% CI, 0.62–1.38; and recessive model: OR, 1.10; 95% CI, 0.56–2.14). The lack of noticeable change in the three models may be due to the high between-study heterogeneity.

Sub-group analysis was carried out in all three models to assess *GPX4* (rs713041) SNP association with pre-eclampsia, separately from ischemic stroke and hypertension. After excluding pre-eclampsia papers from the meta-analysis, the rs713041 T allele was found to be associated with an increased risk of developing stroke and hypertension in comparison with the rs713041 C allele, in the additive model (OR, 6.85; 95% CI, 1.97–23.75; *p* = 0.002), in the dominant model (OR, 1.93; 95% CI, 1.23–3.04; *p* = 0.004) and also in the recessive model (OR, 5.80; 95% CI, 1.43–23.56; *p* = 0.01). On the other hand, when stroke and hypertension studies were excluded from the meta-analysis, the rs713041 T allele was associated with a decreased risk of developing pre-eclampsia in all three models (additive model: OR, 0.68; 95% CI, 0.57–0.82; *p* < 0.0001, dominant model: OR, 0.80; 95% CI, 0.70–0.92; *p* = 0.002; and recessive model: OR, 0.75; 95% CI, 0.64–0.88; *p* = 0.0005) than those with the C allele.

Pre-eclampsia data were further analyzed as they reported data from mild and severe cases and pre-eclampsia early-onset and late-onset. All three models showed a significant decrease in the OR for all the sub-types of pre-eclampsia in the subjects carrying the T allele of *GPX4* (rs713041) SNP with the exception of late-onset pre-eclampsia (dominant model) where there was no difference (*p* = 0.12) (Appendix A).

### 3.4. Meta-Analysis Reproduction-Related Diseases

In an analysis of diseases related to the female reproductive system, 513 cases and 443 controls were included in the initial meta-analysis. Three studies were used: one report evaluated endometriosis [22], one miscarriage [42] and one pregnancy loss [43]. Regarding demographics and genotyping methods, the endometriosis report was performed on the Han Chinese population using TaqMan assay [22] and the miscarriage and pregnancy loss on Russian women using PCR-RFLP [42,43]. Figure 4 shows the meta-analysis results for reproduction-related diseases and the association with *GPX4* (rs713041) SNP. The heterogeneity test was higher than 50% in the three genetic models (I^2^ = 77%, 81% and 54%, respectively), and random-effects analysis was used to explore the influence of rs713041 on reproductive disorders. No significant differences were observed in the additive model (Figure 4A: OR, 0.87; 95% CI, 0.36–2.12; *p* = 0.76), dominant model (Figure 4B: OR, 0.78; 95% CI, 0.40–1.53; *p* = 0.47) or recessive model (Figure 4C: OR 1.08; 95% CI, 0.62–1.86; *p* = 0.79). By excluding one paper at a time to eliminate possible bias in the meta-analysis, no association between *GPX4* (rs713041) SNP and reproductive disorders were observed in all genotype models. No asymmetry was noted in the resultant funnel plots for the additive, dominant and recessive model (Appendix A) and was supported by Egger’s test (additive model *p* = 0.33; dominant model *p* = 0.36; recessive model *p* = 0.30), suggesting the lack of publication bias.

### 3.5. Link of GPX4 (rs713041) Genotype to Other Diseases

Previously, the *GPX4* (rs713041) genotype has been investigated in other diseases, but the number of publications was limited and not sufficient to be included in a meta-analysis. The *GPX4* (rs713041) SNP had been traced in autoimmune thyroid diseases in the Chinese population: Xiao et al. (2017) stated that the major C allele and CC genotype decreased in both types of autoimmune thyroid diseases (Graves’ diseases and Hashimoto’s thyroiditis) patients compared to controls, but no association of this SNP with both diseases was found [23].

Ściskalska et al. (2022) traced the *GPX4* (rs713041) genotype in acute pancreatitis patients in comparison to healthy control [44]. The frequencies of the rs713041 genotypes were similar in healthy subjects, whereas the CT genotype was the most common in the group of acute pancreatitis patients. They also suggested that the TT genotype for rs713041 was associated with an increased risk of acute pancreatitis, possibly contributing to disturbances in the neutralization of oxidative stress [44].

Moreover, the *GPX4* (rs713041) genotype has been investigated in neurological diseases such as Alzheimer’s disease and depression. Da Rocha et al. reported a significant association between *GPX4* (rs713041) SNP and long-term visual memory in Alzheimer’s disease patients, as the T homozygote genotype was more frequent in subjects without long-term memory deficits [24]. Winger et al. (2018) reported an association between *GPX4* (rs713041) SNP and depression as the TT genotype and the T allele increased the risk of depression, whereas CT heterozygous and C allele diminished its risk [25].

Gusti et al. (2021) investigated the association between *GPX4* (rs713041) SNP and type 2 diabetes mellitus and stated that there were no statistically significant differences in genotype distribution between type 2 diabetes mellitus patients and control, while higher levels of triglyceride were reported in *GPX4* (rs713041) C/C cohorts, followed by C/T patients and T/T carriers [45].

Du et al. (2012) investigated the association between *GPX4* (rs713041) SNP in Kashin-Beck disease, an endemic joint disease mainly distributed in areas with low selenium intake, and stated that no significant differences were observed in either genotype or allele frequency between healthy and Kashin-Beck disease subjects [46].

Wigner et al. (2022) assessed the association between *GPX4* (rs713041) SNP and the development of multiple sclerosis (MS) in a Polish population [26]. They detected that the C/C and C allele of rs713041 decreased the risk of MS, whereas the T/T genotype and T allele increased the risk suggesting some links between polymorphic variability in oxidative-stress-related genes and the risk of MS development in the Polish population [26].

### 3.6. Selenium Status and Cancer Risk

Lower selenium levels in plasma or serum have been reported to be closely related to the development of various cancers [48]. Data for selenium levels for both cases and control were extracted from the papers, and values were available from 828 cases and 1287 controls, all of the European origins. The standardized mean difference (SMD) between cases and controls was analyzed via meta-analysis to assess the association between cancer risk and selenium status. As shown in Figure 5, selenium levels were lower in cancer patients compared to control (SMD = −0.39 µg/L; 95% CI: −0.64, −0.14; *p* = 0.002). Sensitivity analysis showed that none of the studies reversed the association between cancer risk and lower selenium status. Publication bias was assessed by visual examination of the funnel plot (Appendix A), and Egger’s test confirmed no publication bias (*p* = 0.34).

Another marker of selenium status is the level of glutathione peroxidase 3 (GPX3) activity in plasma or serum, which is crucial to maintain redox status. Reduced levels in plasma have been associated with increased oxidative stress. Data for GPX3 activity were available in only two reports [17,37], including values for 384 cases and 675 controls. SMD between cases and controls was analyzed to evaluate the association between cancer risk and GPX3 activity (Figure 6). No significant differences were observed in this analysis (SMD = −0.07 U/L; 95% CI: −0.20, 0.05; *p* = 0.25).

### 3.7. Discussion

A systematic review and meta-analysis were conducted to investigate the influence of *GPX4* (rs713041) SNP on susceptibility to several human diseases. This meta-analysis showed a statistically significant relationship between *GPX4* (rs713041) SNP and susceptibility to colorectal cancer and hypertension-related diseases (stroke and hypertension), with carriage of the T allele giving increased susceptibility. On the other hand, the T allele was associated with a decrease in the risk of pre-eclampsia.

The selenoprotein GPX4 contains selenium as part of the active centre and catalyzes the conversion of lipid peroxides from the oxidation of glutathione, which is constantly reused by glutathione reductase (GR) [4]. GPX4 has a distinctive capacity to protect cells from ferroptosis by reducing lipid peroxidation and regulating oxidative stress [49]. It has been suggested that rs713041 SNP can influence GPX4 levels and function. For example, individuals of different genotypes exhibited significant differences in the levels of lymphocyte 5-lipoxygenase total products [50]. In addition, Crosley et al. found that the genotype for *GPX4* (rs713041) SNP in human endothelial cells affected cell function—homologous TT cells were more susceptible to oxidative stress and monocyte adhesion in comparison to CC cells [51].

Due to the potential importance of GPX4 function in several molecular pathways, as recently reviewed by Ursini and colleagues [3], this systematic review focused on determining the association of *GPX4* (rs713041) T/C variation with several diseases. Genotype and allele frequencies in case and control groups were investigated in three different genetic models: the additive model, which assumes a linear and uniform increase based on the number of each copy of the disease-causing allele (T) (the risk for CT is k then the risk for TT is 2k); the dominant model which assumes that having one or more copies of the disease allele (in this study considered as T) increases the risk compared to C (TT or CT genotypes have higher risk); and the recessive model which assumes that two copies of T allele are required to alter the risk (individuals with the genotype TT are compared to individuals having genotypes CT and CC) [52].

*GPX4* (rs713041) genotype T allele was associated with an increased risk of developing colorectal cancer in both additive and dominant models, as well as an increased risk of developing stroke and hypertension in additive, dominant and recessive models. The increased risk for colorectal cancer, stroke and hypertension in the rs713041 allele T genotype may be the result of an effect of the SNP on mRNA translation and, consequently, GPX4 activity and increases in oxidative stress, ferroptosis, and inflammation. Such changes could be expected to impact cancer risk. The data analysis indicates associations between colorectal cancer risk and rs713041 genotype and suboptimal selenium status. This is compatible with increased colorectal cancer risk with lower selenium status in a large pan-European population [53]. Unfortunately, analysis of rs713041 was not possible in this population for technical reasons [54]. However, the central role of selenium in GPX4 function makes it likely that a combination of *GPX4* (rs713041) genotype and selenium status is important in determining disease risk.

The *GPX4* (rs713041) SNP has been associated with the hyperactivation of signaling pathways that generate oxidative stress and inflammation mediators, such as adhesion molecules [51]. TT genotype increased lipid peroxidation, monocyte adhesion and VCAM-1 (vascular cell adhesion protein 1) expression in endothelial cells, being a risk factor for the development of cardiovascular disease [51]. Pre-eclampsia patients experience periods of ischemia and placental reperfusion, generating a hypoxic environment which contributes to the formation of ROS, lipid peroxidation and endothelial dysfunction [55]. Surprisingly, in this meta-analysis, the *GPX4* (rs713041) T allele was associated with a decreased risk of developing pre-eclampsia in all the genotype models. It is important to highlight that all the studies reporting pre-eclampsia data are by Chen et al. and Peng et al., which relate to the Chinese Han population, and all cases and controls were recruited from Shandong Province in China [38,39]. Therefore, these results may be related to specific geographic locations, ethnic groups or environmental factors such as diet, stress, smoking, and micronutrient intake. In fact, serum selenium level in the Shandong Province of China was reported as 120.7 ug/L on average, and the proposed reference range is 39.5–197.4 ug/L [56]. Altogether, the relation between *GPX4* (rs713041) SNP, selenium status and pre-eclampsia needs more studies from different geographic areas to provide a clear understanding.

This study had some limitations which should be considered. In general, diseases with similar characteristics were clustered for meta-analysis, but not all selected studies were included as they did not fit in a specific disease group or there was only one study per disease, which prevented meta-analysis (e.g., depression, Alzheimer’s, diabetes). Another limitation of the study is that the effect of ethnicity on disease risk could not be assessed due to the small number of included studies on different ethnic groups. Finally, a thorough analysis of a combination of genotype and selenium status was not possible. Despite the small number of studies per disease group, the quality of the studies included was high, and publication bias was not present (except in hypertension-related disease meta-analysis), providing strength to the conclusions.

## 4. Conclusions

This is the first meta-analysis to assess the association of *GPX4* (rs713041) SNP with a variety of diseases. Overall, our findings suggest that *GPX4* (rs713041) SNP influences colorectal cancer development and progression with low selenium status playing also a role in different cancers, and that *GPX4* (rs713041) SNP influences the risk of stroke and hypertension. This study, taken together with animal and cell culture work [13,50,51,57], highlights the potential importance of rs713041 in human disease. Future work to better understand how this variant affects human physiology and disease is likely to benefit from the development of transgenic mouse models.

## Figures and Tables

**Figure 1 ijms-23-15762-f001:**
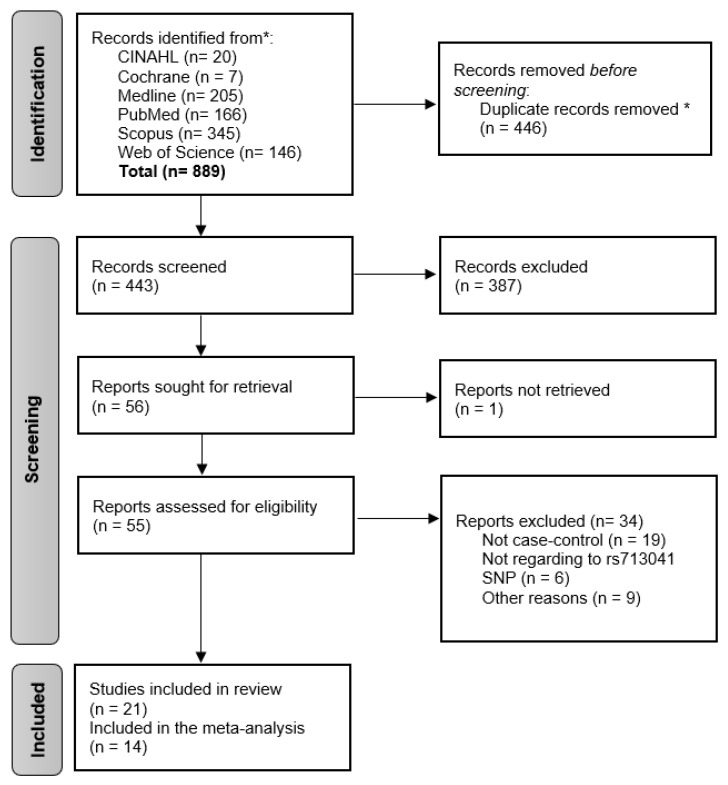
Flow-diagram showing the study selection procedure. * Using Zotero.

**Figure 2 ijms-23-15762-f002:**
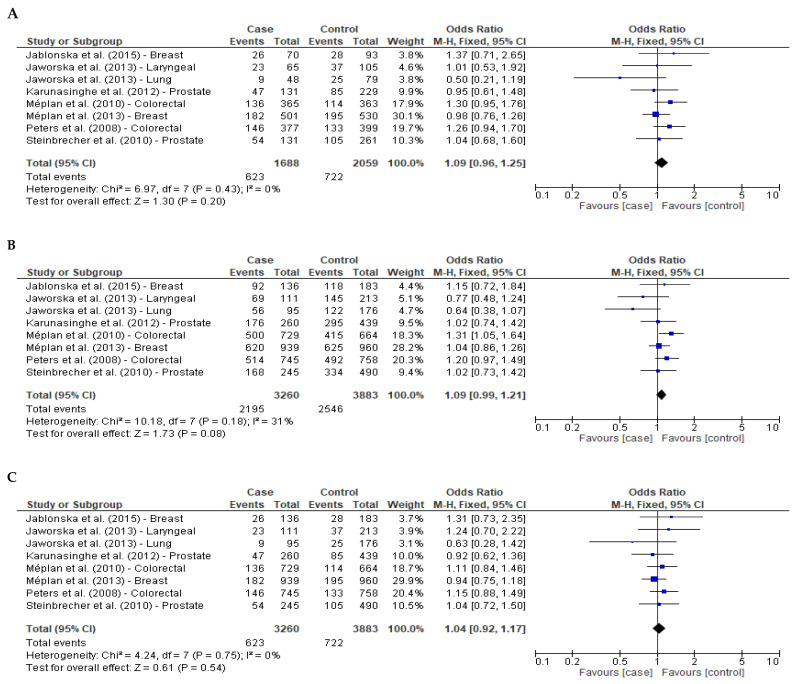
Forest plots of odds ratios with 95% confidence intervals of the association between *GPX4* (rs713041) SNP and risk of cancer. (**A**)—Additive model (TT vs. CC) [16,17,19,20,35,36,37], (**B**)—Dominant model (CT+TT vs. CC) [16,17,19,20,35,36,37] and (**C**)—Recessive model (TT vs. CC+CT) [16,17,19,20,35,36,37].

**Figure 3 ijms-23-15762-f003:**
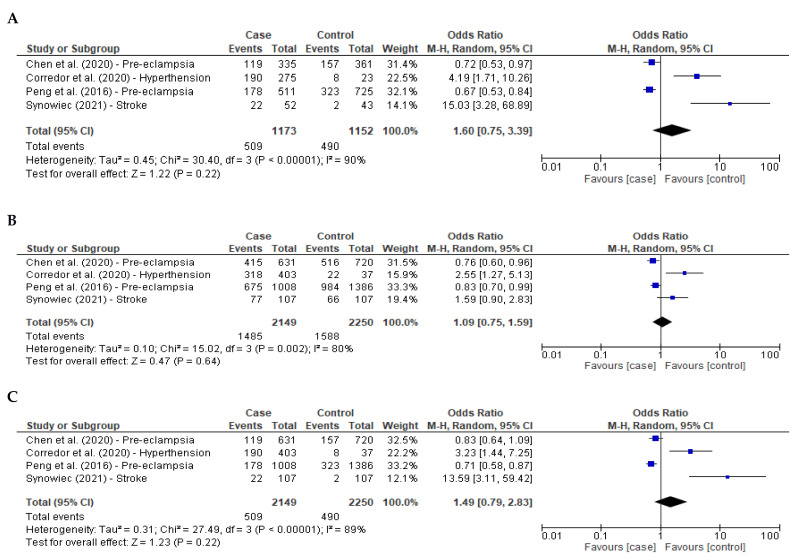
Forest plots of odds ratios with 95% confidence intervals of the association between *GPX4* (rs713041) SNP and risk of hypertension-related diseases. (**A**)—Additive model (TT vs. CC) [38,39,40,41], (**B**)—Dominant model (CT+TT vs. CC) [38,39,40,41] and (**C**)—Recessive model (TT vs. CC+CT) [38,39,40,41].

**Figure 4 ijms-23-15762-f004:**
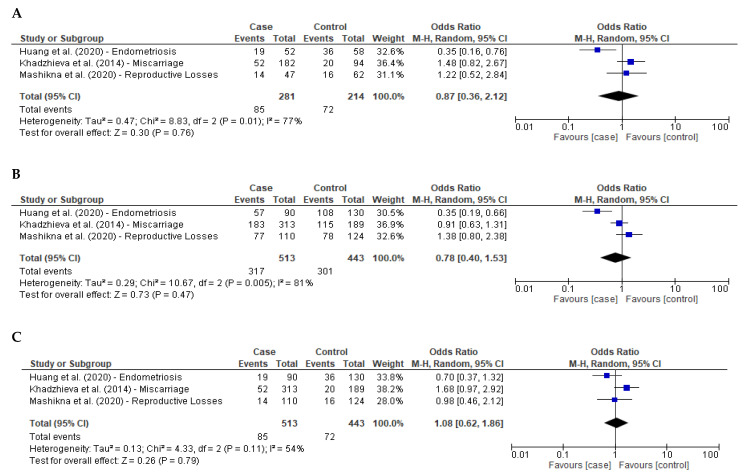
Forest plots of odds ratios with 95% confidence intervals of the association between GPX4 (rs713041) SNP and risk of reproduction-related diseases. (**A**)—Additive model (TT vs. CC) [22,42,43], (**B**)—Dominant model (CT+TT vs. CC) [22,42,43] and (**C**)—Recessive model (TT vs. CC+CT) [22,42,43].

**Figure 5 ijms-23-15762-f005:**
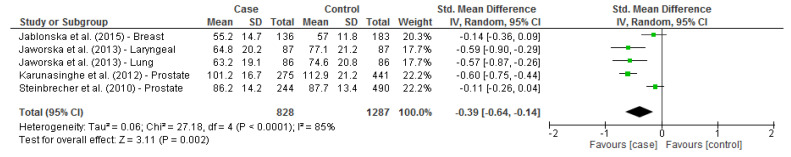
Forest plot of standardized mean difference (SMD) of selenium levels between cases and controls [17,19,20,37].

**Figure 6 ijms-23-15762-f006:**
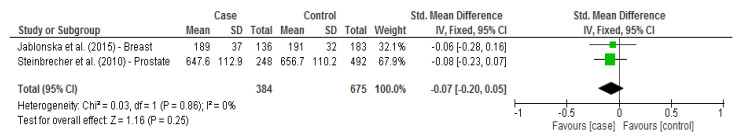
Forest plot of standardized mean difference (SMD) of GPX3 activity between cancer patients and control group [17,37].

**Table 1 ijms-23-15762-t001:** Main characteristics of studies included in the systematic review and meta-analysis (n = 21).

Author (Year)	Country	Ethnicity	Disease	Genotyping Methods	Sample Size (Case/Control)	CC		CT		TT		Newcastle-Ottawa Scale
						Case	Control	Case	Control	Case	Control	
Peters (2008) [36]	USA	>93% Caucasian American	Advanced distal colorectal adenoma	SNPlex or TaqMan	745/758	231	266	368	359	146	133	High
Méplan (2010) [35]	Czech Republic	Czech origin adults >29 years	Colorectal cancer	KASPar	729/664	229	249	364	301	136	114	Moderate
Steinbrecher (2010) [37]	Germany	European men	Prostate cancer	MassARRAY system	245/490	77	156	114	229	54	105	High
Du (2012) [46]	China	Chinese Han population	Kashin-Beck disease	PCR-RFLP	216/194	68	67	124	102	24	25	High
Karunasinghe (2012) [20]	New Zealand	European men	Prostate cancer	TaqMan	260/439	84	144	129	210	47	85	High
Jaworska (2013) [19]	Poland	Polish adults	Laryngeal cancer	Real-time PCR	111/213	42	68	46	108	23	37	High
			Lung cancer		95/176	39	54	47	97	9	25	
Méplan (2013) [16]	Denmark	Danish women	Breast cancer	TaqMan	939/960	319	335	438	430	182	195	High
Khadzhieva (2014) [42]	Russia	Russian women	Recurrent miscarriage	PCR-RFLP	313/189	130	74	131	95	52	20	Moderate
Jablonska (2015) [17]	Poland	Polish women	Breast cancer	TaqMan	136/183	44	65	66 ^#^	90 ^#^	26	28	High
Peng (2016) [39]	China	Chinese Han women	Pre-eclampsia (PE)	TaqMan	1008/1386	333	402	497	661	178	323	Moderate
			Mild PE		158/1386	64	402	70	661	24	323	
			Severe PE		850/1386	269	402	427	661	154	323	
			Early onset PE		523/1386	181	402	253	661	89	323	
			Late onset PE		485/1386	152	402	244	661	89	323	
Xiao (2017) [23]	China	Han Chinese	Thyroid diseases	MassARRAY system	1022/898	318	300	704 *	598 *			Moderate
			Graves’ disease		675/898	213	300	462 *	598 *			
			Hashimoto’s thyroiditis		347/898	105	300	242 *	598 *			
da Rocha (2018) [24]	Brazil	South Brazilian adults	Alzheimer’s disease	TaqMan	103/108	28	34	49	42	25	29	Moderate
Wigner (2018) [25]	Poland	Adults	Depression	TaqMan	281/229	87	83	141	138	53	8	Moderate
Chen (2020) [38]	China	Chinese Han women	Pre-eclampsia (PE)	TaqMan	631/720	216	204	296	359	119	157	Moderate
			Mild PE		141/720	47	204	66	359	28	157	
			Severe PE		490/720	169	204	230	359	91	157	
			Early onset PE		249/720	94	204	112	359	43	157	
			Late onset PE		382/720	122	204	184	359	76	157	
Corredor (2020) [40]	Spain	Spanish	Hypertension	TaqMan	403/37	85	15	128	14	190	8	Moderate
Huang (2020) [22]	China	Han Chinese women	Endometriosis	TaqMan	90/130	33	22	38	72	19	36	Moderate
Mashkina (2020) [43]	Russia	Russian women	Pregnancy loss	PCR-RFLP	110/124	33	46	63	62	14	16	Moderate
Gusti (2021) [45]	Saudi Arabia	Saudi adult population	Type 2 diabetes mellitus	TaqMan	109/163	27	42	62	87	20	34	High
Synowiec (2021) [41]	Poland	Caucasian	Ischemic stroke	TaqMan	107/107	30	41	55	64	22	2	Moderate
Ściskalska (2022) [44]	Poland	Adults	Acute pancreatitis	PCR-RFLP	39/51	9	16	19	18	11	17	High
Wigner (2022) [26]	Poland	Native Polish adults	Multiple sclerosis	Real-time PCR	142/140	39	57	72	65	31	18	High

^#^ values were calculated from CT+TT values reported in the study; * the values represents CT+TT.

**Table 2 ijms-23-15762-t002:** Selenium and GPX3 activity levels reported in this systematic review and meta-analysis.

Author (Year)	Ethnicity	Disease	Sample Size (Case/Control)	Selenium Status (Mean ± SD—µg/L)		Sample Size (Case/Control)	GPX3 Activity (Mean ± SD—Units/L)	
				Case	Control		Case	Control
Steinbrecher (2010) [37]	European men	Prostate cancer	244/490	86.2 ± 14.2	87.7 ± 13.4	248/492	647.6 ± 112.9	656.7± 110.2
Karunasinghe (2012) [20]	European men	Prostate cancer	275/441	101.2 ± 16.7	112.9 ± 21.2	-	-	-
Jaworska (2013) [19]	Polish adults	Laryngeal cancer	87/87	64.8 ± 20.2	77.1 ± 21.2	-	-	-
		Lung cancer	86/86	63.2 ± 19.1	74.6 ± 20.8	-	-	-
Jablonska (2015) [17]	Polish women	Breast cancer	136/183	55.2 ± 14.7	57.0 ± 11.8	136/183	189 ± 37	191 ± 32

**Table 3 ijms-23-15762-t003:** Results of meta-analysis for *GPX4* (rs713041) SNP and individual cancers.

Cancer Type	Cases/Controls	Additive Model (TT vs. CC)		Dominant Model (CT+TT vs. CC)		Recessive Model (TT vs. CC+CT)	
		OR (95% CI)	*p*	OR (95% CI)	*p*	OR (95% CI)	*p*
Breast	1075/1143	1.02 (0.81–1.30)	0.85	1.06 (0.89–1.26)	0.54	0.98 (0.80–1.21)	0.88
Prostate	505/929	1.00 (0.73–1.36)	0.98	1.02 (0.81–1.29)	0.86	0.98 (0.75–1.28)	0.88
Colorectal	1474/1422	1.28 (1.04–1.58)	0.02	1.25 (1.07–1.46)	0.004	1.13 (0.93–1.36)	0.22

Bold denotes significant *p*-value.

## Data Availability

Not applicable.

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
