# Peer review of "The Role of rs713041 Glutathione Peroxidase 4 (GPX4) Single Nucleotide Polymorphism on Disease Susceptibility in Humans: A Systematic Review and Meta-Analysis"

_ijms, 2022, doi:10.3390/ijms232415762_

Round 1

Reviewer 1 Report

Comment 1: It is not clear what is the need for this meta-analysis? Mentioning "no previous meta-analysis" alone is not enough justification. Authors should elaborate more in the introduction about the urgency to do this review. 

Comment 2: The authors repeatedly mentioning that a minimum of 2 studies is enough to conduct a meta-analysis. There is no limited number to conduct a meta-analysis. However, it is about the strength of the conclusion and how confident the results that you presented to the scientific and clinical community. In light of this, no doubt a meta-analysis of minimum of two or three studies is considered immature. 

Comment 3: The introduction focused on the SNP and overlooked the selenium levels and its role in the carcinogenesis, hypertension related or reproductive disorders. Add at least one sentence explaining the importance of selenium to the group of the disease under study. 

Comment 4: The first sentence in the introduction is incomplete it states that " Glutathione peroxidases (GPXs) are enzymes that convert hydrogen peroxide (H2O2) or organic peroxides through the  ....." in this sentence we don't know GPXs converts hydrogen peroxide to what!

comment 5: Methodology suffers from serious short comes:

1.Search strategy is not clear and not following PICOS strategy. I suggest adding a detailed search strategy for each database as a supplementary material.

2. The targeted databases were searched in February 2022! I suggest to update the search, perhaps more studies can be retrieved. 

3. Authors didn't assess the methodological quality for the included studies. Accordingly, quality level of the inferences generated from this study is not known.  If the authors did quality assessment, this is should be mentioned clearly in the methods. 

4. Inclusion criteria didn't include the cancer stage and severity. This is important for the clinical implication of the investigated SNP. 

5. Data extraction regarding the selenium levels not mentioned at all in the methods yet shown up in the results. With regards to the selenium further information is needed like methods of determination of selenium, are the studied patients under treatment or not?, cancer staging and severity. 

6. Reporting bias not assessed at all in any of the investigated categories.  

7. Calculation of standardized mean difference for the selenium is not explained (cases from controls or vice versa). 

8. PRISMA checklist is absent, it should be added as a supplementary material.

Comment 6: The result didn't reflect the analysis plan. The authors need to explain in the statistical analysis all steps and decisions made during the analysis process. For instance, page 9, line 223-225, it is not clear what authors meant by this "Since significant heterogeneity existed in 
all three models, pre-eclampsia, ischemic stroke, and hypertension were evaluated separately" did authors mean sub-group analysis ??! or something else?! What is done should be described in the statistical analysis. 

Comment 7: Data regarding the selenium levels is missing entirely, neither in the tables nor in supplementary materials! it should be shown in the manuscript. 

Comment 8: in page 11 subheading 3.6. selenium status and cancer risk. authors use the term Standard mean difference (SMD). Please correct to standardized mean difference. 

Author Response

Reviewer 1

We thank the reviewer for the comments. We believe the changes we have made to the manuscript (as also per suggestion of reviewer #2) will improve our manuscript and highlight the importance of this meta-analysis and its contribution to existing knowledge. Throughout the manuscript, we have highlighted in yellow the sections we have changed.

Comment 1: It is not clear what is the need for this meta-analysis? Mentioning "no previous meta-analysis" alone is not enough justification. Authors should elaborate more in the introduction about the urgency to do this review. 

We have re-written most of the introduction to highlight the importance to carry out this systematic review and the link of with selenium status

Comment 2: The authors repeatedly mentioning that a minimum of 2 studies is enough to conduct a meta-analysis. There is no limited number to conduct a meta-analysis. However, it is about the strength of the conclusion and how confident the results that you presented to the scientific and clinical community. In light of this, no doubt a meta-analysis of minimum of two or three studies is considered immature. 

We appreciate that the conclusions drawn might have limitations but we believe it is important to analyse such studies and report finding. Moreover, the quality of studies include in the meta-analysis has been assessed (see comment 5, point 3) and the all the manuscript are of high/moderate quality (score>7 according to Newcastle-Ottawa scoring methods for case-control studies). Publication bias has been also tested.

Comment 3: The introduction focused on the SNP and overlooked the selenium levels and its role in the carcinogenesis, hypertension related or reproductive disorders. Add at least one sentence explaining the importance of selenium to the group of the disease under study. 

As per comment 1, we have re-written the introduction and highlighted the importance of selenium status in various diseases

Comment 4: The first sentence in the introduction is incomplete it states that " Glutathione peroxidases (GPXs) are enzymes that convert hydrogen peroxide (H2O2) or organic peroxides through the  ....." in this sentence we don't know GPXs converts hydrogen peroxide to what!

We have changed part of the introduction and the above sentence is not present anymore

Comment 5: Methodology suffers from serious short comes:

1.Search strategy is not clear and not following PICOS strategy. I suggest adding a detailed search strategy for each database as a supplementary material.

We have now created a table to summarise the string used to search the different databases and we have also considered the PICOS strategy.

  1. The targeted databases were searched in February 2022! I suggest to update the search, perhaps more studies can be retrieved. 

We have carried out new searches to address the gap between February and November 2022. Four more studies were identified, only one met the inclusion criteria and it is now included in Table 1. Of the 3 that did not meet the inclusion criteria, 2 did not report genotype data and one did but the quality of the data was not good. Part of the manuscript made reference to T/C genotype for GPX4 rs713041 whereas in other parts it was reported as A/T

  1. Authors didn't assess the methodological quality for the included studies. Accordingly, quality level of the inferences generated from this study is not known.  If the authors did quality assessment, this is should be mentioned clearly in the methods. 

We have carried quality assessment for all the studies included in the manuscript using the Newcastle-Ottawa scale for case control studies and all the studies were classified of high or moderate quality.

  1. Inclusion criteria didn't include the cancer stage and severity. This is important for the clinical implication of the investigated SNP. 

We do agree with the reviewer that the cancer stage and severity of disease is very important for the potential clinical implications. Unfortunately, it was not possible to select papers for cancer stage or severity as the information related to genotype was not present in function of disease stage.

  1. Data extraction regarding the selenium levels not mentioned at all in the methods yet shown up in the results. With regards to the selenium further information is needed like methods of determination of selenium, are the studied patients under treatment or not?, cancer staging and severity. 

The data information about selenium levels and GPX3 activity have been included in the methods section. Similarly to the previous comment, the information related to cancer staging and severity were not present in the papers.

  1. Reporting bias not assessed at all in any of the investigated categories.  

We have assessed publication bias by visual analysis of funnel plots and by carrying out Egger’s test. Funnel plots are now reported next to each meta-analysis and results from Egger’s test are reported and discussed in the text.

  1. Calculation of standardized mean difference for the selenium is not explained (cases from controls or vice versa).

This has now been clarified in the methods section. It has been calculated as cases from controls.  

  1. PRISMA checklist is absent, it should be added as a supplementary material.

We have now included PRISMA checklist as supplementary Table S1

Comment 6: The result didn't reflect the analysis plan. The authors need to explain in the statistical analysis all steps and decisions made during the analysis process. For instance, page 9, line 223-225, it is not clear what authors meant by this "Since significant heterogeneity existed in 
all three models, pre-eclampsia, ischemic stroke, and hypertension were evaluated separately" did authors mean sub-group analysis ??! or something else?! What is done should be described in the statistical analysis. 

We agree with the reviewer that the section statistical analysis was nto comprehensive. We have nto changed to data analysis and we have included all the analysis we have carried out with the manuscript

Comment 7: Data regarding the selenium levels is missing entirely, neither in the tables nor in supplementary materials! it should be shown in the manuscript.

We apologise for the oversight. We have now included a new table (Table 3) with the data regarding selenium levels and GPX3 activity  

Comment 8: in page 11 subheading 3.6. selenium status and cancer risk. authors use the term Standard mean difference (SMD). Please correct to standardized mean difference. 

We have now corrected the wording

Reviewer 2 Report

The paper entitled: “The role of rs713041 Glutathione Peroxidase 4 (GPX4) single

nucleotide polymorphism on disease susceptibility in humans:

a systematic review and meta-analysis” is clearly presented and well-written.

Authors have carried out a meta-analysis of the association between GPX4

18 (rs713041) SNP and various diseases in human, such ascancer, hypertension, and stroke.

I have few minor comments:

• C allele is written both with and without hyphen, also there are allele T and allele C; I would choose one way of writing these words. • Figure 1 text is hard to read.  • Line 167 “two breast cancer” misses “for”.

Author Response

Reviewer 2

The paper entitled: “The role of rs713041 Glutathione Peroxidase 4 (GPX4) single nucleotide polymorphism on disease susceptibility in humans: a systematic review and meta-analysis” is clearly presented and well-written.

Authors have carried out a meta-analysis of the association between GPX4  (rs713041) SNP and various diseases in human, such as cancer, hypertension, and stroke.

 We would like to thank you the reviewer for the constructive comments on our manuscript. Please find below response to your comments. Throughout the manuscript we have highlighted in yellow the changes we have made

Comments:

  • C allele is written both with and without hyphen, also there are allele T and allele C; I would choose one way of writing these words. 

We have now made it uniform and choose the for C allele or T allele

  • Figure 1 text is hard to read.  

We have updated Figure 1 and made it bigger

  • Line 167 “two breast cancer” misses “for”.

Thank you for spotting this. We have now added the word ‘for’

Round 2

Reviewer 1 Report

I would like to thank the authors for addressing and responding to the previous comments. The manuscript has improved markedly. I have some minor/major comments need to be address before accepting this manuscript

Major

1.     In the "Meta-analysis Hypertension-related diseases" as there is evidence for publication bias please perform trim-and fill method to investigate whether the potentially missed articles would change the effect estimate or not. And please comment on the generated result accordingly.

2.     In the abstract material and methods please remove the search keywords and write the statistical analysis namely the effect size measured and the software used.

3.     In the abstract write the selenium SMD, 95% CI , p-value and I2 values.

Minor

1.     Please write the p-value in 3 decimal points instead of 2 throughout the paper.

2.     Please move all funnel plots to the supplementary files.

Author Response

We want to thank you for your time and effort to review the manuscript.

Reviewer 1

I would like to thank the authors for addressing and responding to the previous comments. The manuscript has improved markedly. I have some minor/major comments need to be address before accepting this manuscript

We thank the reviewer for the additional comments, which have improved further our manuscript. Throughout the manuscript, we have highlighted in yellow the sections we have changed.

Major

  1. In the "Meta-analysis Hypertension-related diseases" as there is evidence for publication bias please perform trim-and fill method to investigate whether the potentially missed articles would change the effect estimate or not. And please comment on the generated result accordingly.

When publication bias was present (Meta-analysis Hypertension-related diseases), we have used  the Duval and Tweedie’s trim and fill method to determine where missing studies are likely to fall, and then recalculate the combined effect. The results have been discussed in the text and modified funnel plots included as supplementary Figure S2.

  1. In the abstract material and methods please remove the search keywords and write the statistical analysis namely the effect size measured and the software used.

We have modified the abstract according to the suggestions and highlighted in yellow the changes.

  1. In the abstract write the selenium SMD, 95% CI , p-value and I2 values.

We have now included the above values in the abstract.

 Minor

  1. Please write the p-value in 3 decimal points instead of 2 throughout the paper.

Thank you for the comment but unfortunately Review Manager version 5.4 does not provide 3 decimal point for p values unless the values are P<0.00x. We apologise for not being consistent with the use of decimal points throughout the manuscript but we felt it was not appropriate to re-run all the analyses using a different software.

  1. Please move all funnel plots to the supplementary files.

We have followed the reviewer’s suggestion and we also added the funnel plot from the trim and fill method for hypertension-related diseases (supplementary Figure S2).
